# The Research Progress of Extraction, Purification and Analysis Methods of Phenolic Compounds from Blueberry: A Comprehensive Review

**DOI:** 10.3390/molecules28083610

**Published:** 2023-04-21

**Authors:** Xinyu Bai, Lin Zhou, Li Zhou, Song Cang, Yuhan Liu, Rui Liu, Jie Liu, Xun Feng, Ronghua Fan

**Affiliations:** 1Department of Sanitary Inspection, School of Public Health, Shenyang Medical College, Shenyang 110034, China; 2Department of Food Science, School of Public Health, Shenyang Medical College, Shenyang 110034, China; 3Department of Sanitary Chemistry, School of Public Health, Shenyang Medical College, Shenyang 110034, China

**Keywords:** blueberry, phenolic compounds, extraction, separation, purification, analysis and detection

## Abstract

Blueberry is the source of a variety of bioactive substances, including phenolic compounds, such as anthocyanins, pterostilbene, phenolic acids, etc. Several studies have revealed that polyphenols in blueberry have important bioactivities in maintaining health, such as antioxidant and anti-tumor activities, immune regulation, the prevention of chronic diseases, etc. Therefore, these phenolic compounds in blueberries have been widely used in the field of healthcare, and the extraction, isolation, and purification of phenolic compounds are the prerequisites for their utilization. It is imperative to systematically review the research progress and prospects of phenolic compounds present in blueberries. Herein, the latest progress in the extraction, purification, and analysis of phenolic compounds from blueberries is reviewed, which can in turn provide a foundation for further research and usage of blueberries.

## 1. Introduction

Blueberry (*Vaccinium* spp.) belongs to the family Ericaceae, subfamily Vaccinoideae, genus Vaccinium [1]. It is popular because of its fresh and sweet fruit and rich nutritional value. Recently, the blueberry was called a “super fruit” by the Food and Agriculture Organization of the United Nations (FAO). It is one of the five healthiest fruits in the world. In recent years, blueberry production and consumption have increased with significant economic value. Blueberries are classified into four species based on morphological classification: rabbit-eye blueberry (*V. virgatum*), northern high shrub (*V. corymbosum*), southern high shrub (*V. formosum*), and low shrub (*V. angustifolium*) (See Figure 1).

Most parts of the blueberry plant can be used, including the fruit, leaves, and stems. The fruit is consumed raw or processed into juice, jam, wine, etc.; leaves and stems are the alternative source of bioactive natural products [2]; and its distinctive color is often used as natural dye. Moreover, because of its rich nutritional content, the blueberry has been developed into many nutritional supplements for disease prevention and dietary regulation, such as blueberry lutein eye care tablets, blueberry dietary supplements (such as blueberry powder), pterostilbene capsules, etc. (See Figure 1).

Blueberry fruit is greatly appreciated for its nutritional value of enriched organic acids (citric, ascorbic, phenolic acids, and tannins), phenolic compounds (stilbenoids, tannins, and flavonoid compounds, including anthocyanin, flavanone, flavanol, and quercetin), sugar, minerals, vitamins, fibers, and pectins [3]. Phenolic compounds are divided into endogenous and exogenous phenols as per their source, and those present in plants are called endogenous phenols. By chemical composition, phenolic compounds can be classified into tannins, flavonoids, stilbene compounds and phenolic acids. Blueberries are rich in endogenous phenols, which are the most beneficial components in blueberries and include anthocyanins, tannins, pterostilbene and phenolic acids. Table 1 shows the specific types of phenolic compounds in blueberries. Phenolic compounds are utilized as natural colorants and preservatives in foods [4]. Studies have shown that synergies between phenolic compounds in blueberries may be responsible for their high oxidation potential, and they can thus be employed as nutritional supplements [5]. A daily intake of blueberries (about 1/3 cup) can prevent the onset of chronic diseases, such as obesity, diabetes, atherosclerosis, etc. [6]. This review summarizes the extraction, purification, and detection techniques of phenolic compounds in blueberries.

In fact, 50–80% of the total polyphenols were phenolics (3000 mg/kg), while anthocyanins accounted for 60% (16–160 mg/100 g) [20]. Anthocyanins have antioxidant, eye protection, and cardiovascular protection effects. The latest studies depict that anthocyanins reduce inflammation caused by obesity [21]. The study by Trushina, E.N. et al. on rat gastrocnemius showed that anthocyanins promoted apoptotic cell recovery [22]. Pterostilbene is a polyhydroxystilbene compound produced as a natural plant antitoxin by the plant defense system. Pterostilbene as an antioxidant is orally used to prevent the squamous cell carcinoma of skin [23]. Pterostilbene, as the derivative of resveratrol, has similar chemical characteristics and action mechanisms as resveratrol [24]. It also has analgesic, antiaging, antidiabetic, anti-inflammatory, antiobesity, antioxidant, cholesterol-lowering, and neuroprotective properties [25]. Moreover, it depicts higher biological activity, stability, and bioavailability [26]. Pan, J. et al. found that pterostilbene is the bioactive component of the blueberry, which reduced serum uric acid and protected the kidneys in hyperuricemic nephropathy (HN) mice by reducing the serum creatinine, urea, urinary albumin, and urinary albumin to creatinine ratio (u-ACR) [27].

Tannins can be divided into hydrolyzed tannins (HTs) and condensed tannins (CTs). In addition, these can be divided into five groups according to chemical classification: gallic tannins, tannins, complex tannins, condensed tannins, and luminous tannins [28]. Blueberries are rich in tannins, which has high research significance. Furthermore, as a polyphenol, tannins are mainly found in the vacuoles of plants, and in blueberries tannins are found in leaves, roots, stems, and pericarp. Since the industrial sector is interested in natural extracts, it is important to optimize the process using sustainable extraction methods to obtain higher extract weights and tannins. Therefore, it is necessary to review the extraction, detection, and analysis methods of tannins. 

Phenolic acids have a beneficial impact on human health and are employed to prevent diseases ranging from atherosclerosis to ischemic stroke [29]. Phenolic compounds and total phenolic composite plants are increasingly used in modern neurology practice, such as for treating and preventing cerebral ischemia and neurodegenerative diseases [30,31]. Blueberries could be used as phenolic compounds source in future because of the expending blueberry production, improvement in blueberry yield and innovation in developing various processes. Extraction, purification, detection, and analysis methods of phenolic compounds in the blueberry are being widely explored and researchers have developed diverse new strategies. Phenols (including flavonoids, such as resveratrol, curcumin, coumarin, and phenolic acids) have anti-inflammatory, antioxidant, and neural protection characteristics [32]. Anthocyanin is the most abundant phenolic compound in blueberries followed by phenolic acid. Flavonoids content, such as pterostilbene, was relatively high. Phenolic acids are represented by chlorogenic acid in the blueberry. The functional active units are formed between the monomers through covalent binding and biochemical reactions, which are scattered in the plant. The chemical structures of one major phenolic substance in the blueberry are shown in Figure 2.

Current studies of blueberries are primarily focused on their nutritional and medicinal value, seedling cultivation, and the separation of anthocyanins, pterostilbene, and all kinds of acids. The detection methods are explored for the further utilization of blueberries. Herein, the extraction, separation, and detection methods of phenolic compounds in blueberries from recent years are summarized. They are categorized according to the current research progress. The research status of phenolic compounds in blueberries is encompassed, and their future utilization is suggested.

## 2. Research Progress of Extraction, Purification, and Detection Methods of Some Phenolic Compounds in Blueberries

### 2.1. Anthocyanines

#### 2.1.1. Extraction Methods

The anthocyanin structure consists of two benzene rings linked by the units of omega-3 carbon (C6-C3-C6). The parent nucleus of anthocyanin is 2-phenylbenzopyrane, the floral motif [33]. Anthocyanins protect the heart and nervous systems as they possess antidiabetic, antiobesity, anticancer, and other effects. They also prevent retinal degeneration. In vitro and in vivo epidemiological studies have confirmed the therapeutic effects of anthocyanins [34]. Anthocyanins are not only present in blueberries but also in leaves, pomace, and seeds [35,36]. Blueberries are a good biological source of anthocyanins. There are multiple methods for extracting anthocyanins from the blueberry; however, several factors affect the stability and yield of anthocyanins. It is thus difficult to compare the extraction yields of anthocyanins achieved through various methods. Therefore, this review analyzes the current progress and lists extraction and separation methods with their working conditions in order to improve the methodologies. 

##### Solvent Extraction

The most commonly used extraction method for anthocyanin is solvent extraction, which uses the difference in solubility or the partition coefficient of substance in two immiscible (or slightly soluble) solvents for transferring the substance from one solvent to another. According to anthocyanin solubility, commonly used solvents for extracting anthocyanin are methanol, ethanol, and others. However, methanol is toxic and environmentally unfriendly, and the stability of extracted anthocyanins is not good. Conventional solvent extraction methods may lead to the partial or total hydrolysis of anthocyanin and decrease in its activity. SEM involves convenient operation, simple equipment, and easy implementation; however, it has the disadvantages of long time, low efficiency, large solvent consumption, and high temperature [37]. Lff., A., and Nmm, B. et al. improved the solvent extraction by extracting anthocyanins from blueberry residue in acidic water (1% citric acid). The optimal combination of extraction time and temperature was 5 min (−1.41) and 100 °C. The resulting solution had high concentrations of polyphenols concentration and antioxidant activity. This optimized extraction method minimizes extraction time, used inorganic and safe solvents and did not require advanced equipment for increasing yields [7].

##### Ultrasound-Assisted Extraction

UAE is a new green extraction method, based on the polarity and solubility of active components, where they enter into the solvent under ultrasonic waves to obtain multi-component mixed extract. The extract is then separated, refined, and purified to finally achieve the desired chemical components. UAE can extract anthocyanins from blueberry residue. The optimal extraction conditions were 40 °C, 400 W ultrasonic power and 40 min extraction time and obtained the best yield of 108.23 mg/100 gDW [35]. Moreover, multi-frequency ultrasonic extraction can be carried out at the optimal extraction conditions: dual-frequency ultrasound 40 + 80 kHz, ultrasonic power 350 W, extraction temperature 50 °C, and an extraction time of 40 min. Compared with single-frequency (25 kHz), the anthocyanin yield through three-frequency (25 + 40 + 80 kHz) and dual-frequency ultrasounds was increased by 15.26% and 5.45%, respectively. The antioxidant activity (DPPH, hydroxyl radical scavenging ability and reducing power) of anthocyanins extracted by dual-frequency ultrasound was higher than that extracted without ultrasound [8]. Aliaño-González, M.J., and others improved the conditions of ultrasonic-assisted extraction under 0 °C with 5 mL and containing 34.20% MeOH 74 mL of solvent extraction in the aqueous solution of pH 6—best conditions were seen when the sample was 4.70 g and the best time was 25 min. Compared with magnetic stirring i, more anthocyanins were extracted by applying ultrasound (5.9% at 16 min and 60.3% at 42 min). It was demonstrated that more anthocyanins could be extracted by employing ultrasound for short time under optimal extraction conditions [9]. Under prolonged and increased power ultrasound, the ·OH concentration was increased and anthocyanin activity was decreased. The results revealed that cavitation was the main mechanism of anthocyanin degradation. Cavitation assisted in ·OH production from water molecules, which led to the ring-opening degradation of anthocyanins. The blueberry anthocyanins degradation followed first-order reaction kinetics, and the storage time of blueberry anthocyanins was the longest at room temperature in 20% ethanol solution [38].

##### Microwave-Assisted Extraction (MAE)

MAE uses microwave energy to extract substances. It is a method for extracting chemical components from plants or animal tissues by using suitable solvents in a microwave reactor. Xue et al. applied MAE to extract anthocyanin from blueberry. They studied the impact of microwave power on extraction rate. With microwave power of 100 W/g, the extract temperature was 53~58 °C and anthocyanin yield was the highest [10].

##### Extraction Using Deep Eutectic Solvent

DES is a class of green solvents commonly associated with ionic liquids because of common properties, such as high thermal stability, low volatility, and low vapor pressure [39]. Natural deep eutectic solvents (NADES) are composed of natural compounds produced by cell metabolism, and they have similar characteristics to DES. Crude blueberry extracts based on NADES exhibited increased anthocyanins bioavailability compared to organic solvent extracts [11], Grillo, G. et al. demonstrated that US and MW could provide a broad enhance extraction efficiency compared to conventional extraction procedures [40]. A study based on freeze–melt technology revealed that FUTE could quickly and efficiently extract ABVS and maintain its antioxidant potential with higher extraction rates [41]. PEF treatment is a non-thermal method that uses direct current (DC). The electric potential passes through living cell membranes, opens protein channels and increases cell permeability. Arruda, H.S. et al. found that for maximum anthocyanin extraction yield, the best PEF treatment was to use low or moderate intensity PEF treatment, thus avoiding its degradation [42]. These new green extraction technologies can extract anthocyanins from blueberry fruits and apply them to the recycling of kitchen waste. Future applications include anthocyanin recovery and use of these compounds in diverse fields [43].

##### Enzyme-Assisted Extraction

The principle of enzyme-assisted extraction is to use the high specificity of enzymes to treat plant materials whose cell walls are not easy to directly extract through solvents. The process changes the permeability of cell walls dominated by the cellulose. The pectin in the plant is completely decomposed into small molecules, which improves the extraction rate of active components. Enzyme-assisted extraction is one of the few methods that achieves the extraction of complex phenolic compounds. It provides mild conditions, substrate specificity, and industrial applicability [44]. Granato, D. et al. compared common enzymes and found cellulase as the most efficient enzyme to recover anthocyanins and other phenols and improves antioxidant activity [12]. Presently, there are fewer applications of anthocyanins enzymatic extractions from blueberries, despite having broad development potential.

##### Extraction Using Supercritical Fluid Carbon Dioxide

As a new extraction technology that integrates extraction and separation, supercritical fluid extraction has simple extraction conditions, the protection of active components, less reagent consumption, a fast rate and almost no solvent residue. CO_2_ is the most commonly used fluid. Qin, G.W. et al. improved the SCDE method with optimized conditions: extraction temperature 40 °C, pressure 34.7 MPa, CO_2_ flow rate 4.5 L/min, extraction time 1.86 h and yield 1.48 mg/g. However, this method required a high pressure environment, expensive equipment, and high maintenance costs, which limited its large scale usage [13].

##### Combined Extraction Method

Jin, Y. et al. adopted an improved homogenization–ultrasonic-assisted extraction (HUAE) method by adding carnosic acid as a natural antioxidant. This method ensured anthocyanins and flavonols yields with effective antioxidant roles in the extraction process [45].

UAE is a fast and efficient method for recovering phenols. The increase in yield through UAE is attributed to acoustic cavitation, which involves the formation, growth, and collapse of microbubbles on solid surface. This leads to corrosion and erosion and, finally, the breakdown of the cell wall, which allows the solvent to penetrate the solid and enhances mass transfer. Pressurized liquid extraction (PLE) is performed using liquid solvent at high temperatures and pressures. UAE + PLE uses a mixture of hydroethanol (50% water and 70% ethanol *v*/*v*) to produce an extract with a higher content of phenolic compounds and antioxidant capacity. PLE + UAE in a water-alcohol solvent improves antioxidant extraction compared to PLE and UAE individually [14].

Jovanović, M.S. et al. extracted blueberry anthocyanins using natural deep eutectic solvent (NADES) combined with UAE. The results showed that choline chloride:sorbitol (1:1) was the most efficient extraction solvent. Thecyanoglyco-3-o-glucoside yield was 0.2751 mg/g DW and the TAC yield was 2.12 mg CGE/g under optimal extraction conditions (37.63 min, 48.38 °C, 34.79% (*w*/*w*) water in NADES) [15]. DES extracted polyphenols, carbohydrates and lipids and could be employed for food safety monitoring and biosensor development as a feasible alternative. Regarding the present study, more work is required to resolve these shortcomings and determine whether compounds extracted through DES are safe to eat [46]. Further studies are needed for anthocyanin extraction from blueberry in order to develop quick, efficient, and environmentally friendly methods in which bioactivity and yield are ensured. The integration of extraction methods for phenolic compounds from the blueberry is given in Table 1.

#### 2.1.2. Separation and Purification

Anthocyanins and pterostilbene are important components of polyphenols. However, the crude anthocyanin extract often contains starch, pectin, and other impurities. The product has low purity and poor stability which should be separated and purified. The common separation and purification methods of anthocyanins include resin purification, chromatography, liquid phase extraction, solid phase extraction, etc. Resin purification is the most commonly used method. Presently, various green purification methods, such as high speed countercurrent chromatography, extraction based on compressed fluid and whirlpool-assisted dispersion–liquid microextraction, are used for the extraction and purification of phenolic compounds.

##### High Performance Liquid Chromatography (HPLC)

HPLC is often used for anthocyanin separation. A study has shown a HPLC method where six major anthocyanins are simultaneously separated, with 99% purity and 22.5% yield by using acetonitrile–water (containing 0.3% phosphoric acid) as mobile phase gradient elution at 520 nm detection wavelength [47].

Wang, E. et al., separated three pure monomeric anthocyanins-3-O-glucoside and morning glory-3-O-glucoside by semi-preparative HPLC and identified them by HPLC-DAD-ESI-MS/MS. These anthocyanins purities were determined by HPLC as 97.7%, 99.3% and 95.4%, respectively. This indicates that fractionation purity by semi-preparative chromatography was higher than anthocyanins separation by HPLC [48].

High-speed counter-current chromatography (HSCCC) uses a kind of liquid stationary phase and liquid mobile phase liquid–liquid chromatographic separation. It utilizes a two-phase solvent system to establish unidirectional hydrodynamic equilibrium within high-speed spinning solenoid, a technology that has been widely used for natural product separation and is well suited for the efficient large-scale separation of polyphenols and their derivatives in order to achieve high purity of up to hundreds of milligrams within hours per run [49]. Degenhardt, A. et al. found that the duplexic mixture of tert-butyl methyl ether/n-butanol/acetonitrile/water (2:2:1:5) acidified with trifluoroacetic acid is a suitable solvent system for anthocyanin separation [49]. However, because of the large capital investment, it could not be applied to large-scale industrial applications. RP-LC method is also a common method for anthocyanin separation and purification; however, it was difficult to determine a unified standard due to varying experimental conditions.

##### Column Chromatography

The principle of column chromatography is to take advantage of the distribution coefficients of anthocyanins in the solid phase and mobile phase to better separate anthocyanins and impurities [50]. Xue, H. et al. found that AB-8 macroporous resin had the highest adsorption rate of blueberry anthocyanins, i.e., 97.73%. The desorption rate was the highest at 81.52% (adsorption conditions: 1.0 mL/min for feed flow, 1.0 mg/mL for anthocyanin concentration, pH3.0; desorption conditions: eluate flow rate 1.5 mL/min, ethanol concentration 60%, pH 3.0). Moreover, after one treatment of macroporous resin combined with Sephadex LH-20, the anthocyanins purity increased from 4.58% to 90.96%, an increase of 19.86-fold [51]. Yao, L. et al. found that polymer sorbents with EGDMA content of 20% (DE-20) exhibited the best adsorption capacity and selectivity for anthocyanins compared to various commercial adsorbents [52]. This novel highly acidic cation exchange resin 001X7 was employed in wastewater treatment and for anthocyanins separation from mulberry extracts using a large column system. This method had been demonstrated through small scale and large scale experiments to achieve concentrations above 95% at a lower cost and in less time than MAR and SCX [53].

##### Membrane Separation

Membrane separation refers to selectively separating the mixtures of the molecules of diverse sizes at the molecular level as they pass through the semi-permeable membrane. Current membrane separations and coarse extractions include microfiltration (MF) membranes, ultrafiltration (UF) membranes and nanofiltration (NF) membranes. Alexandru, M., and Avram et al. optimized NF membrane separation for achieving a volume reduction of more than 60% without significantly reducing the biological activity of anthocyanins [54].

##### Progress in Coupled Separation Techniques

Chorfa, N. et al. isolated anthocyanin molecules using hydrophobic silica gel (DSC-C18) and cation exchange resin (DSC-SCX) through two consecutive solid phase extraction (SPE) and HPLC processes with 100% purity [55]. Vilkickyte, G. et al. isolated fifteen compounds from bilberry, including major and minor anthocyanins and their glycosides, in 20 min using HPLC-PDA, as well as nine compounds from cranberries and twelve compounds from bilberry [56].

#### 2.1.3. Detection and Analysis Methods

##### Ultraviolet-Visible (UV-VIS) Spectrophotometry

UV-VIS is based on the absorption characteristics of molecules in the UV-VIS electromagnetic radiations ranges. The degree of absorption with other characteristics establishes a qualitative and quantitative analysis method. Today, the official international analysis method of the Association of Official Analytical Chemists (AOAC) for anthocyanins is the pH difference spectrophotometric method, which is a fast direct quantitative method (AOAC, 2005) [57]. The UV-VIS absorption spectrum of anthocyanins shows strong absorption in visible range at about 520 nm. However, it does not provide enough structural information to identify single anthocyanin in complex samples. The analysis method is prone to error detection because of two or more anthocyanins in chromatographic proximity [58].

##### Chromatographic Method

HPLC uses liquid chromatography to quantitatively and detect anthocyanins in samples. Myjavcová, R., and others found that methanol medium, when used as a mobile phase for carrying out fractionation in the stationary phase, obtained suitable chromatographic performance by employing less polar solvent (i.e., ethanol and ethyl). It resulted in phase compression and provided repeatable results [59].

##### Mass Spectrometry

Mass detection provides higher sensitivity, lower LOD and detailed structural information as compared with LC-DAD. Anthocyanin analysis commonly employs quadrupole MS, triple quadrupole (QQQ) MS and quadrupole-time-of-flight (QTOF) MS, etc. However, Q-TOF and ESI are the most commonly used analyzer and ionization method, respectively [58].

##### Coupled Detection Methods

DAD + MS or tandem mass spectrometry (MS/MS) are often used for anthocyanin detection in practical applications. Longo, L. et al. used HPLC-DAD in conjunction with MS to determine anthocyanins in berries. They extracted pigments from berry peels with 0.1% HCl methanol, purified using C-18 solid phase columns, and identified by HPLC-DAD-MS. Information from the HPLC mapping, saponification and acid hydrolysis of anthocyanins showed that the main anthocyanins were geranolin 3-O-rutinoside (64%), geranium 3-O-glucoside (16%) and geranium 3-O-trans p-coumanoside (13%) [60]. Kalogiouri, N.P. et al. modified the HPLC-DAD method for determining anthocyanins in the skin of Greek red grapes [61]. Myjavcová, R., also used LC-MS and MS2 methods in L. caerulea var. Several less common pigments were found in Kamteschatica fruit. Based on precise mass measurements and spectral studies, 5-methylpyranothocyanins, cyanidin-3-hexoside dimers and compounds containing anthocyanin and catechin units linked by ethyl bridges were first identified in Vacciniaceae and Caprifoliaceae berry extracts [59]. Due to the shortcomings of LC-MS and UV spectrophotometry, Saha, S. et al. employed LC-MS in combination with UV-VIS spectroscopy to provide additional information on the structural details of anthocyanins. If analytical reference standards were available, the 34-pole mass spectrometer provided selectivity and quantitative sensitivity in the analysis. Moreover, high resolution mass spectrometry (HR-MS) in the screening of non-target compounds in the preliminary evaluation process provided valuable information in the absence of reference standards [62]. Thuy, N.M. et al. used the UPLC–UV–MSin combination with LC–ESI–QQQ (6460 Triple Quadrupole System, Agilent, St. Clara, CA, USA) and further coupled with UV detector (1260 Infinity, Agilent, St. Clara, CA, USA) for determining anthocyanins and their derivatives in butterfly pea flowers through the analytical column Zorbaz Eclipse C18 (2.1 × 50.0 mm, 1.8 μm, Agilent, St. Clara, CA, USA). The mobile phase consisted of water (solvent A) and acetonitrile (solvent B), each containing 0.1% formic acid. This method provided the reliable determination of anthocyanins at a flow rate of 0.3 mL/min [62]. Combining HPLC–DAD, mass spectrometry (MSn) and electrospray ionization (ESI) data, Gavrilova, V. et al. developed methods and characterized the phenolic properties of blueberries, red currants and blackcurrants grown in Macedonia and identified the individual derivatives of phenolic acids, flavonols, flavonol-3-alcohols and anthocyanins by UV and mass spectrometry. These were quantified using LC–DAD. Finally, data revealing the diversity and quantity of phenolic compounds in these fruits were obtained [63]. Moreover, according to Prior, R.L. et al., the HPLC–MS combination was also a useful method to distinguish anthocyanins from proanthocyanidins [64].

### 2.2. Pterostilbene

#### 2.2.1. Extraction Methods

PTS is one of the most abundant compounds in blueberries and is a natural analogue of resveratrol, which is mainly found in blueberries. PTS has a series of pharmacological properties, especially anti-inflammatory and anticancer. Pterostilbene has higher stability for intestinal permeability and absorption than resveratrol fat because its chemical structure exists in two methoxyl forms. Moreover, PTS has less toxicity and fewer adverse effects when used in cancer chemoprevention and chemotherapy applications [65]. PTS is a candidate for treating a variety of diseases, including diabetes, cancer, cardiovascular neurodegenerative, and aging, because of its higher bioavailability and lower toxicity compared to other astragalus [66]. PTS is combined with anthocyanins in blueberries and is somewhat inconvenient to extract. A Chinese patent (application number CN201510107787.6) first used acidic water to extract anthocyanins by reducing the interference from PTS. The authors employed ultra-high pressure extraction equipment to extract PTS. Devgun, M. et al. determined that MAE could be used to extract anthocyanins from PTS and compared the traditional and emerging extraction methods [16]. However, current studies have confirmed that high pressure can cause protein folding and denaturing, and whether the method of extracting red sandalwood astragalus from blueberries under ultra-high pressure destroys the protein in blueberries has not yet been studied [67]. At present, synthetic pterostilbene is mostly used, and no further studies on PTS extraction from blueberries are known.

#### 2.2.2. Separation and Purification

PTS has strong hydrophobicity and is combined with cyclodextrins 4–6 to increase its solubility in water. Cyclodextrins 4–6-based micellar electrokinetic capillary chromatography is more suitable for the separation and purification of pterostilbene than conventional chromatography; however, overall efficiency is not high and complexation efficiency is low. Catenacci, L. et al. used cyclodextrin to bind PTS and achieved the complete dissolution of PTS in 40 and 20 min through β-CD and γ-CD, respectively. Waszczuk, M. et al. identified PTS:BCD:HPMC ternary system using ethanol as auxiliary solvent. This yielded low-volume powders with high PTS content, providing an alternative to producing solid or semisolid formulations containing highly soluble PTS [17,68]. The methods of separation and purification of phenolic compounds from blueberries are summarized in Table 2.

#### 2.2.3. Detection and Analysis Method Mass Spectrometry (MS)

Mass spectrometry (MS) is a method that uses electric and magnetic fields to separate moving ions according to their mass-to-charge ratio and then detects the compounds. Through the analysis of the molecular weight of these ions, the compounds, chemical structures, and cracking rules can be obtained. This method has the characteristics of high accuracy and high sensitivity, which are especially suitable for separation of mixtures. Becker, L. et al. used MALDI–MS imaging to simultaneously localize resveratrol, pterostilbene, and glucoside on grape leaves [70]. Because MALDI is a soft ionization technique, molecular information can be retained and there is no need to label the compounds of interest for detection, as with fluorescence microscopy. Therefore, it provides a method of “label free” imaging. This is convenient and effective for the localization detection of pterostilbene.

##### Chromatographic Method

HPLC is a common method for PTS detection in plants. It is used to find the differences in the annual pterostilbene content of the same species for determining the optimal variety [71]. Waszczuk, M. et al. discovered a specific stability indicator, PTSβCD, for the quantitative analysis of pterostilbene stilbene in dried blueberries through HPLC. The method was reliable with relatively high accuracy and sensitivity [72]. Rodríguez-Bonilla, P. et al. developed a reverse-phase high performance liquid chromatography (RP–HPLC) method for determining pterostilbene in food samples. The method was based on the addition of cyclodextrins (CDs) to the mobile phase, wherein pterostilbene was complexed by CDs. Studies demonstrated that the addition of 0 mM HP-β-CD to the 25:7 (*v*/*v*) methanol aqueous mobile phase at 25 °C and pH 7.0 improved the main analytical parameters [73].

##### Coupled Detection Methods

The LC–MS method is commonly used to detect pterostilbene concentration in rat plasma. The coupled method combines the separation power of chromatography with the qualitative function of MS, usually using ether as solvent to extract plasma samples. Rodríguez-Cabo, T. et al. used, for the first time, dispersion liquid–liquid microextraction (DLLME) to determine the concentration of three hydroxylated stilbenes (trans isomers of pterostilbene, resveratrol, and averatrol) in wine samples. The method was as follows: the acetylation step was performed prior to DLLME, followed by gas chromatography mass spectrometry (GC–MS) analysis to reach a limit of quantification (LOQ) of 0.6 to 5 ng/mL (−1). The results exhibited a linear response of up to 5000 ng/mL (−1) and excellent accuracy (recoveries ranged from 90–102% for samples spiked from 50 to 1000 ng/mL (−1)). Sample injection and organic solvent consumption were maintained at 1 and 0.6 mL, respectively. The total sample preparation (derivatization and concentration) time was 15 min [74]. Mazzotti, F. et al. proposed a new and reliable method for PTS determination in various matrices through LC–MS/MS and isotopic dilution. The method sensitivity was demonstrated based on the selectivity of LC–ESI, the specificity of MS/MS, and the accuracy of isotopic dilution method [75]. Xie, L. et al. characterized stilbene in California almonds by UHPLC–MS and identified stilbene polyphenols [76]. There are many identification methods of pterostilbene in organisms (such as pterostilbene content in plasma [77]); however, few detection studies of pterostilbene in blueberries are available.

### 2.3. Phenolic Acids

#### 2.3.1. Extraction Methods

Phenolic acids are also the important components of blueberry phenolic compounds in addition to anthocyanins. The phenolic acids in blueberries include chlorogenic acid (CGA), ferulic acid, cinnamic acid, etc. Each 100 g of blueberries contain 85 mg phenolic acids [78]. Of these, CGA is the main functional component of phenolic acids in blueberries, generated by caffeic acid and quinic acid (1-hydroxyhexahydrogallic acid). CGA has a strong antioxidant effect, which removes free radicals in human cells and reduces oxidative damage. It also has antibacterial properties, prevents cardiovascular control diseases, improves blood pressure, and stimulates the central nervous system and other functions [79].

##### PEF Law

Ante Lončarić et al. found that in ethanol-based solvents, phenolic acid extraction was highest at 15 min (442.90 μg/g DW) with 50 Hz, while UAE was more efficient for methanol-based solvents after 26 min at 15 °C (80.561 μg/g DW). Compared with the other two methods, the highest TPC (100.20 μg/g DW) was determined in the extracts obtained with PEF-assisted extraction in ethanol after 625 pulses and 47 kV/cm. In conclusion, PEF improved the extraction rate of phenolic acids from blueberry pomace and was thus a reliable green extraction method [80].

##### Ultrasound-Assisted Solvent Extraction (UASE)

UASE is low cost with high separation efficiency. Studies have revealed the advantages of using UASE to isolate CGA from plants at laboratory and industrial scales. Compared to the conventional extraction methods, UASE processing time is shorter, the cost is lower, and the extraction efficiency is higher. It prevents the thermal degradation of compounds and guarantees a larger contact area between the solid and liquid phases [18].

##### Coupled Extraction Techniques

In a study on CGA recycling from agricultural waste, the authors compared MAE, UAE, and CSE methods for recovering CGA from blueberry leaves and found that MAE was the most effective [81]. Wang, T. et al. combined a deep eutectic solvent with aqueous two-phase system to integrate and sustainably separate CGA from blueberry leaves [19]. Xie, L. et al. used aqueous two-phase extraction to recover CGA from plant leaves [82].

In addition to the extraction methods, new synthesis methods not only broaden the utilization channels of phenolic acids but also lay a foundation for further utilization. For example, phenolic acid-g-chitosan has stronger antioxidant and anti-tumor ability than chitosan. It can also be applied to biological packaging materials or adsorbent materials, which has broad economic prospects. Phenolic acid-g-chitosan can be synthesized using carbodiimide group coupling, enzyme-catalyzed grafting, free radical-mediated grafting, and electrochemical methods [83].

#### 2.3.2. Separation and Purification

##### Resin Adsorption Method

Macroporous resin, silica gel, dextran gel resin, ion exchange resin, etc., are used for the separation and purification of phenolic acids. However, at present, there are few such studies regarding phenolic acids in blueberries. The research scope is wide as blueberries are an important source of phenolic acids.

##### Chromatographic Method

In addition to high-speed countercurrent chromatography, Jiang, H. et al. developed an analytical strategy for separating phenolic acids by ultra-high performance convergence chromatography (UPC). Studies have shown the BEH column to be the best method. An efficient separation in minimum time of 0.8 min was achieved through gradient elution with carbon dioxide and methanol/acetonitrile (70:30, *v*/*v*) at a flow rate 0.8 mL/min and with 17% TFA as modifier [69]. The pH zone refining countercurrent chromatography had been employed in the separation of natural and synthetic mixtures, especially the organic acids and alkaloids [84]. Ma, T. et al. isolated 94 phenolic acids, including citric acid from echinacea using pH zone-refining countercurrent chromatography [85].

#### 2.3.3. Detection and Analysis Methods

##### Chromatographic Method

GC requires high temperature treatment, and thus the sample may decompose at high temperatures. GC is not often used for the detection of phenolic acid compounds. However, GC can be employed to separate and analyze small phenolic acid compounds below 600 Da. GC-MS is a useful hyphenated method of the analyzing phenolic acids in plant samples. Compared with LC-MS, GC-MS has higher selectivity, precision, and accuracy, especially when analyzing components of lesser concentrations. The detection methods for phenolic compounds in blueberries are summarized in Table 3.

HPLC is utilized for the quantitative analysis of phenolic acids in plants. The strategy depends on the chemical characteristics of pre-detection components, extraction methods, particle size, storage time, and conditions. Moreover, the determination method and interferents (such as fats, terpenes, and chlorophyll) are important. The components are quantified by HPLC, as it is the most often used quantitative detection method. Chen, H. et al. employed magnetically assisted tube solid phase microextraction to adsorb and desorb phenolic acids from fruit juice prior to HPLC detection. The process increased the analyte extraction efficiency from 44.9–64.0% to 78.6–87.1%. Under optimal extraction conditions, the detection limit was 0.012–0.061 μg/L, the relative standard deviation of intra-day and inter-day variations were 1.9–9.8% [86]. In HPLC methods, phenolic acids are detected using diverse columns, mobile phases, column temperatures, and flow rates. Water, methanol, and acetonitrile are the most common constituents of mobile phase systems, and modifiers, such as formic acid, ammonium acetate, and acetic acid, are sometimes added to prevent peak tailing. Moreover, reversed-phase high performance liquid chromatography (RP-HPLC) also detects phenolic acids, and Padilha, C.V. et al. used RP-HPLC to determine phenolic acids in Brazilian grape juice, which had good linearity and precision and was used for commercial characterizations [87].

##### Capillary Electrophoresis (CE)

CE is widely used in analyzing phenolic acid compounds. CE separates and detects ionic, non-ionic, polar, and non-polar compounds. CE has the advantages of having a small electrolyte volume, short analysis time, high resolution, and small volume sample. For higher resolution, parameters such as buffer, pH, concentration, capillary type, electrophoretic temperature, voltage, and injection method are optimized. de Souza Campos Junior, F.A. et al. extracted phenolic acids from edible fungi based on acid-hydrolyzed water and analyzed extracts by CE-DAD. The optimal conditions for extracting phenolic acid compounds from mushroom were identified by the central composite design: hydrochloric acid 2 mol·L^−1^; temperature, 80 °C; and time, 30 min. This method reduced the impact on environment and can thus be used as a green extraction and detection method [88].

##### Coupled Detection Techniques

Liu, S. et al. utilized UHPLC-DAD-ESI-QTOF-MS and UHPLC-DAD to quantify and characterize phenolic compounds other than anthocyanins in blueberry juice and fermented wine. These methods detected dicaffeoylquinic acid, larch 3-O-galactoside, isorhamnetin 3-O-galactoside, and (+)-catechins in blueberry juice and wine [89]. Furthermore, Sun, J. et al. established a HPLC-evaporative light scattering detection method (HPLC-ELSD) to simultaneously determine 11 phenolic acids and 12 triterpenes in red stilbene, which improved detection efficiency, accuracy, and effectiveness [90].

### 2.4. Tannins

#### 2.4.1. Extraction Methods

##### Solid–Liquid Extraction (SLE)

The principle of SLE is to use solvents to dissolve tannins in cells, and commonly used solvents include water, ionic solvents, organic solvents, etc. A Soxhlet device is often used to extract the tannins dissolved in water and organic solvent [91]. Hagerman found that season, temperature, solvent, and leaves combined with different treatments all affected tannin extraction [92]. Durgawale, T.P. et al, found a combination of the following methods to extract tannins from wood and bark: the Folin–Ciocalteu method for total phenols, vanillin (with sulfuric acid), or HCl/BuOH/Fe (Ⅱ) depolymerizing proanthocyanidins and the Ate–Smith method for ellagitannins (or degradation in MeOH/HCl, followed by ellagic acid if the ellagitannin content is too low) [93]. Compared with this method, newer extraction methods, such as ionic solvent methods, are comparatively more efficient. Ionic liquids are mostly liquid organic salts. This kind of solvent has low vapor pressure, strong thermal stability and can dissolve a large number of substances, which greatly reduces the extraction time. However, ionic solvents are too expensive to use on an industrial scale. Due to the high cost of ionic solvents, water is used in industry as the preferred solvent for SLE. Water can extract tannins at concentrations between 29 and 887 mg/g and produces residues without serious negative effects to the environment. The tannin recovery rate of this solvent extraction is also very high, and the only disadvantage is that a large amount of solvent is required [28].

##### Supercritical Fluid Extraction (SFE)

Similarly to the SFE that we mentioned earlier, carbon dioxide can also be used as solvent. CO_2_ is a non-polar molecule, whereas tannins are considered polar compounds. The actual proportion of “solvent-soluble” and “solvent-resistant” tannins depends on the nature and concentration of the organic solvent [92]. Therefore, in order to improve the solubility of tannins and their extraction efficiency, polar co-solvents, such as ethanol, methanol, and water-based mixtures, are often added. This method of using CO_2_ as a solvent can create a non-toxic anaerobic environment. It can avoid strong oxidation reactions, and relatively mild temperatures do not destabilize tannins. The disadvantage is the high cost of using high-pressure equipment, which is not suitable for large-scale industrial applications. The continuous mixing of solvent and particles also aids the extraction process, and better mixing results can be obtained when the ratio of solvent to solid is higher. However, this ratio can change the particles but not the solvent of tannin extraction [94].

##### Microwave-Assisted Extraction (MAE)

The method is a combination of traditional tannin extraction solvent and rapid microwave heating. According to the principle of solvent and material balance, the greater the amount of solvent, the greater the amount of tannin-10 extraction. In the work of Wang et al., it was found that an increase in solvent volume caused the extracted tannins to strengthen to a certain extent (ratio 1:30) and then begin to decline [95]. The main advantages of MAE are that compared with traditional solid–liquid extraction, the extraction time is greatly shortened and the solvent requirements are reduced. Another advantage is that it improves mass transfer. The main disadvantages are the high cost of equipment required for mass production and the potential thermal degradation of raw materials and tannins [96].

##### Pressurized Water Extraction (PWE)

The method is based on the mass transfer properties of water to tannin at high temperatures and pressures. The main difference from traditional SLE with hot water is that pressurized water extraction uses acceptable temperatures above the boiling point and pressures above atmospheric pressure to keep water in a liquid state. Tannins are extracted under subcritical conditions (0.1–22 MPa, 100–374 °C) most of the time, but pressure and temperature also affect extraction. For example, when subcritical conditions are exceeded, extraction time will be shortened by 5–60 min [91]. Therefore, we can selectively extract different polar tannins by changing the temperature, pressure, or solvent. However, due to the high costs, it is not suitable for large-scale industrial production.

##### Ultrasound-Assisted Extraction (UAE)

Ana Rita Silva et al. compared HAE and UAE to determine which method was most suitable for promoting L from *Cytinus hypocistis* (L.), the subspecies of the genus *Chrysanthus*. UAE has been shown to have some advantages over HAE for intracellular tannin extraction, such as a shortened extraction time. In their study, the authors found that while lower ethanol concentrations produced higher extraction rates (probably due to a higher recovery of water-soluble carbohydrates), the highest levels of tannins were obtained using higher ethanol concentrations. In addition, the authors demonstrated that ethanol/water mixtures (60:40, *v*/*v*) were superior to other organic solvents [97].

#### 2.4.2. Separation and Purification

Tannins from crude plant extracts can be separated and purified by adsorption chromatography. Dan Liu et al. further isolated and purified hydrolyzed tannins from geranium crude extract by adsorption chromatography on a cross-linked 12% agarose gel (Superose 1210/300 GL). The purities of geranyl and geranyl extracted using this method were 92.4% and 87.2%, respectively, and the corresponding yields were 88.0 and 76.8% [98]. In addition, high speed countercurrent chromatography is often used to separate these tannins. Dan Liu et al. purified gallic acid, cilantro, and geranin from 70% ruthenium–iridium acetone aqueous extract by positive and reverse-phase high speed countercurrent chromatography. The solvent system used in this study was n-hexane:ethyl acetate:methanol-acetic acid:water (1:10:0:0:2:20). The results show that tannin can be successfully separated by reverse-phase high speed counter-current chromatography and normal-phase high speed counter-current chromatography [99].

#### 2.4.3. Detection and Analysis

Chen Wang et al. used UHPLC-Q Exactive Orbitrap/MS positive and negative modes, respectively, under optimized conditions (optimized conditions: ethanol concentration (pH = 3), 48%; temperature, 50 °C; and static cycle times, 3. number of static cycles) to identify different types of polyphenols in blueberry extract. Eleven pro-anthocyanidins (tannins) in blueberry were identified by this method, including gallocatechin and epigallocatechin [100]. Annalisa Romani et al. analyzed and characterized several tannin extracts of myrtle and pomegranate using the HPLC/DAD/ESI-MS method and confirmed that water and water ethanol myrtle leaf extracts were rich in gallic side, gallic quinic acid, ellagitannins, and flavonoids. The specific operating conditions of the mass spectrometer are as follows: gas temperature, 350 °C; flow rate, 10.0 L/min; atomizer pressure, 30 psi; tetrode temperature, 30 °C; and capillary voltage, 3500 V. Monopolyphenols were quantified directly by HPLC/DAD using a five-point regression curve established by available criteria [101]. In addition, the tannin content of blueberries can be determined by ultraviolet spectrophotometry. The specific method is as follows: the 1:50 diluted sample and blank sample were hydrolyzed with acid and the absorbance was measured at 550 nm. The resulting absorbance (sample blank) was then multiplied by a factor of 19.33 to calculate the total tannin concentration in g/L. These final values are expressed in mg/100 g dry matter (mg/100 g per minute) [102].

## 3. Outlook

### 3.1. Food Value and Prospects of Blueberries

Blueberry fruits and their derivatives are appreciated by consumers because of their rich bioactive substances, health benefits, and unique flavor [103]. Blueberries are eaten directly or consumed through drinks and foods such as blueberry juice, blueberry wine, blueberry powder, blueberry jam, and dried blueberries [104]. Blueberry fruit and blueberry pomace are also used for processing. They are of nutritional value and provide fiber and bioactive compounds. They can be added to bakery products for additional health benefits [105]. However, they may affect the taste of baked products. A study on producers’ use of blueberry pomace as a fiber ingredient added to biscuits revealed that consumers pay more attention to food deliciousness compared to its health benefits. So, future research and development regarding blueberry food should be focused on both health and taste [106]. These drinks and foods have various health-promoting effects. Blueberry polyphenols cause fluctuations in the pool of small molecule metabolites that are present at the baseline and contribute to human metabolism [107]. A study showed that the frequent consumption of blueberries improved postprandial blood sugar in sedentary individuals and short-term blueberry supplementation improved insulin sensitivity [108]. Hence, consuming blueberries as part of balanced and varied diet has benefits for human health [109]. Furthermore, current research on the interaction of dietary proteins and phenols has immense potential. A study in healthy volunteers confirmed that the combined consumption of milk and blueberry fruit reduced the peak plasma concentration of constitutive phenolic acids, as well as caffeic acid absorption. Studies showed that the benefits of blueberries are better when they are not consumed at the same time as milk. It may be possible to develop new and efficient blueberry foods in future [110].

### 3.2. Health Value and Prospects of Blueberries

Blueberry extract has high economic value in terms of healthcare products. In addition to dietary supplements, existing healthcare products made from blueberry extract, such as anthocyanin/proanthocyanidin, are marketed for their antioxidant and eye protection functions. Anthocyanin mediates relaxation effects on ciliary muscle and blood vessels through the endothelin-B receptor and nitric oxide/cyclic GMP pathway, and thus, the regulation of the lens can ultimately be improved [111]. The daily intake of blueberry fruit in older adults improves cognition, especially the short-term or long-term memory. Stote, K.S. et al. conducted a randomized, single-blind human intervention trial using 100% wild blueberry juice and observed that the short-term consumption of 240 mL of wild blueberry juice per day may promote cardioprotective effects in adults at risk of type 2 diabetes by improving the systolic blood pressure (possibly through nitric oxide production) [112,113]. The berries and stilbene in blueberries also exhibit preventive effects for several cancers [114]. Stull, A.J. et al. employed double-blind and placebo-controlled experiments to study the effect of blueberry phenols on blood pressure regulation and found that adding blueberry powder to the diet improves vascular health and endothelial function in adults with metabolic syndrome [115]. Gagnon, W. et al. reported that the intervention of freeze-dried blueberry powder regulated BAs in feces and enhanced the toxin elimination [116]. Sidorova, Y.V. et al. obtained polyphenol concentrate by adsorbing blueberry extract into ground buckwheat flour. This concentrate had the potential for in-depth research and development as part of specialized foods for preventing digestive diseases, such as metabolic syndrome, m diabetes [117]. The results revealed that blueberry food had a vast range of health benefits, along with a delicious taste and high nutritional value. The development of high-quality food products and supplements can be focused on as a research topic because of the high anthocyanin/proanthocyanidin content in blueberry and grape seed and the related lower extraction costs [118]. Furthermore, the incorporation of blueberry pomace powder into starch chitosan film improves film performance, which is environmentally friendly and degradable and could replace the plastic packaging in future [119]. Combined with the current global blueberry production and planting areas, the extraction, separation, and purification of blueberry phenols has high research potential and feasibility. The blueberry market represents a broad cash crop, where consumption is significant. In the future, the blueberry may become a source of raw material for industrial extracts.

## 4. Conclusions

The blueberry’s potential as a nutrition source and its economic value cannot be undermined. In recent years, progress has been made in extracting phenolic compounds from blueberries by employing conventional solvent extraction methods, such as DES, UAE, MAE, supercritical fluid extraction, and other emerging techniques. Compared with routine extraction strategies, the new extraction methodologies have the advantages of high yield, low energy consumption, short time, environmental protection, etc. There are also some disadvantages, however, such as the high costs and sophisticated equipment requirements. Most industries utilize ultrasonic or microwave extractions. Some new green extraction methodologies cannot be used in large-scale applications because of their escalated costs, which requires an urgent solution. Moreover, the separation and purification techniques for phenolic substances in blueberries have also been improved. In addition to common HPLC separation and purification methods, several coupled extraction techniques have evolved, such as vortex-assisted dispersion-liquid microextraction and solid–liquid extraction. The advantages of these developed methodologies are to limit the need for manpower and material resources and improve yield, and the disadvantages include sophisticated instrument requirements. It has been found that studies are more focused on analyzing and detecting f anthocyanins. The innovative detection methods for pterostilbene and phenolic acids are progressing slowly. The state of research at home and abroad reveals that the main research focus of phenolic compounds in blueberries is concentrated on anthocyanins, and there are few studies on the isolation, purification, and detection of phenolic acids and pterostilbene.

## Figures and Tables

**Figure 1 molecules-28-03610-f001:**
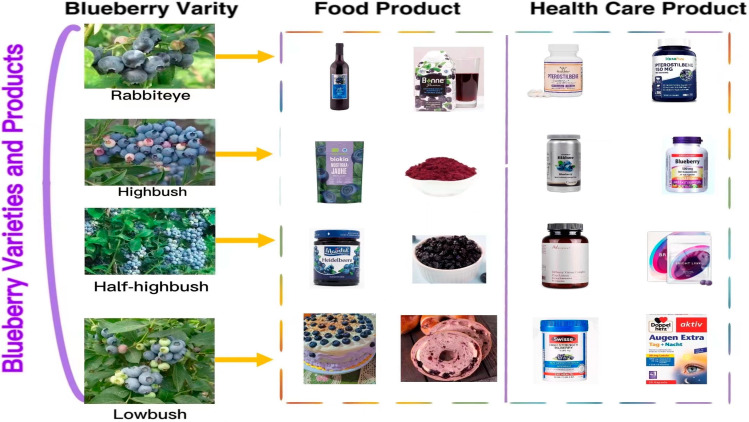
Varieties of blueberries and their products (the pictures of blueberry varieties are from the Fruit Information Center’s wechat public account, the pictures of bread are from the open source website https://pixabay.com accessed on 20 March 2023, and the pictures of food and health products are from the official website of the individual products).

**Figure 2 molecules-28-03610-f002:**
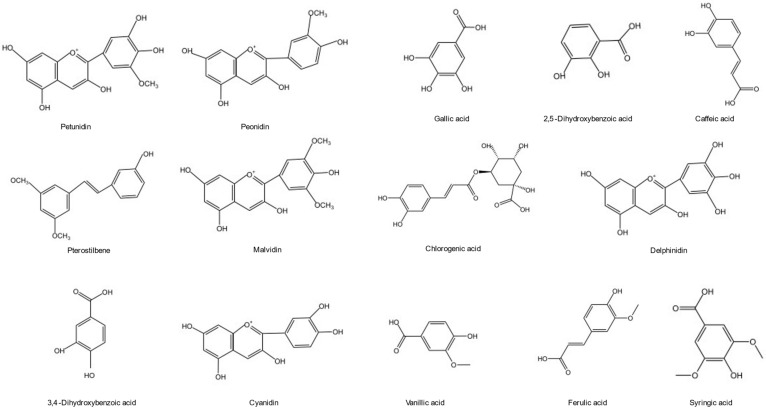
Chemical structure of the main phenols in the blueberry.

**Table 1 molecules-28-03610-t001:** Summary of extraction methods of phenolic compounds from blueberries discussed in this paper.

Phenolic Compound	Extraction Methods	Extraction Condition	Solvents	Ref.
Anthocyanin	SEM	5 min (−1.41) and 100 °C	1% citric acid	[7]
	UAE	1.400 W, 0 °C, PH6, 25 min; dual-frequency: 40 + 80 Hz, 350 W, 50 °C, 40 min	1.34.2% MeOH	[8,9]
	MAE	53–58 °C, 80 s, 100 W/g	60% ethanol solvent	[10]
	DES/NADES	Blueberry powder (8 g; average particle size: 715.5 ± 12.3 μm) was mixed with 20 mL of a methanol:water:formic acid mixture (50:48.5:1.5; *v*/*v*/*v*).	Choline chloride:glycerol:cit-ric acid (0.5:2:0.5, molar ratio) NADES	[11]
	EAE	Cellulase auxiliary extraction	4% acetic acid in 65% aqueous methanol	[12]
	SCDE	Extraction temperature 40 °C, pressure 34.7 MPa, CO_2_ flow rate 4.5 L/min, extraction time 1.86 h,	Carbon dioxide	[13]
	UAE + PLE	Ultrasonic bath for 8 min at 80 °C, PLE for 30 min	50% and 70% ethanol *v*/*v*	[14]
	UAE + NADES	37.63 min, 48.38 °C, 34.79% (*w*/*w*) water in NADES	choline chloride: sorbitol (1:1)	[15]
pterostilbene	MAE	1350 W at 100% power, 30 min	aqueous and ethanolic	[16]
Phenolic acid	PEF	15 min (442.90 μg/gdw) at 50 H	Ethanol-based solvents	[17]
	UASE	80% methanol at room temperature for 15 min	80% methanol	[18]
	MAE	800 W, 50 °C, extraction time from 2 to 5 min	water	[18]
	NADES + ATPE	Liquid/solid ratio 17.01 mL/g, extraction temperature 59.03 °C, extraction time 24.12 min	0.5% (*v*/*v*) formic acid aqueous water (A) and acetonitrile (B)	[19]

**Table 2 molecules-28-03610-t002:** The separation and purification methods of phenolic compounds from blueberries are summarized in this table.

Phenolic Compound	Separation and Purification Methods	Separation and Purification Condition	Purity or Characteristic	Ref.
Anthocyanin	HPLC method	Acetonitrile–water (containing 0.3% phosphoric acid) as the mobile phase gradient elution at 520 nm detection wavelength	99%	[47]
	Semi-preparative high performance liquid chromatography	Mobile phase A: methanol; mobile phase B: 3% formic acid. The initial gradient composition: 15% solvent A and 85% solvent B. The elution conditions: solvent B: 0 min, 85%; 3 min, 80%; 7 min, 75%; 10 min, 75%; 55 min, 30%; 60 min, 30%; 65 min.	The tree anthocyanin components’ purity are 97.7%, 99.3%, and 95.4%	[48]
	HSCCC	Duplexic mixture of tert-butyl methyl ether/n-butanol/acetonitrile/water (2:2:1:5) acidified with trifluoroacetic acid	Peak purity standard	[49]
	Macroporous resin method	Macroporous resin combined with the Sephadex LH-20 method	90.96%	[51]
	Membrane separation method	Nanofiltration membranes (NF245 and NF270) used to separate and adsorb anthocyanins	Reduces anthocyanin waste by more than 60%	[54]
	DSC-C18 + DSC-SCX + HPLC	Hydrophobic silica gel (DSC-C18) and cation exchange resin (DSC-SCX) in two consecutive solid-phase extractions anthocyanins.	100%	[55]
Pterostilbene	New ways to increase solubility	PTS:BCD:HPMC ternary system using ethanol as a co-solvent	Low-bulk powders with a high content of PTS	[17,68]
Phenolic acid	UPC	Mobile phase: carbon dioxide methanol/acetonitrile (00:70, *v/v*) Flow rates of 30.1 mL/min; modifier: 17% TFA; the shortest time: 0.8 min.	A short separation time of 0.8 min	[69]

**Table 3 molecules-28-03610-t003:** Summary of the detection methods of phenolic compounds in blueberries in this paper.

Phenolic Compound	Detection Method	Characteristic	Ref.
Anthocyanin	UV-VIS	Maximum absorption in the visible light range around 520 nm	[58]
	HPLC	Chromatographic fractionation of methanol media achieves suitable chromatographic performance	[59]
	Mass spectrometry	Q-TOF and ESI are the most commonly used methods	[58]
	HPLC-DAD + MSn	Extracts pigments from berry peels with 0.1% HCl methanol, purified using C-18 solid phase columns	[60]
	UV-VIS + LC-MS	Provides additional information on the structural details of anthocyanins	[62]
	UPLC/UV/MS	The mobile phase A: water; mobile phase B: acetonitrile; each containing 0.1% formic acid. With a flow rate of 0.3 mL/min, this method provides a reliable determination of anthocyanins.	[62]
	HPLC + MSn	Distinguishes anthocyanins from pro-anthocyanidins	[64]
Pterostilbene	MALDI mass spectrometry	Simultaneously localizes resveratrol, pterostilbene, and glucoside on grape leaves	[70]
	RP-HPLC	The addition of 0 mM HP-β-CD to a 25:7 (*v*/*v*) methanol aqueous mobile phase at 12 °C and pH 50.50 significantly improved the main analytical parameters	[73]
	DLLME	A linear response: 5000 ng/mL (−1); accuracy: overall recovery of 1000% to 1% for samples spiked at different levels of 90 to 102 ng/mL (−50) with a standard deviation of less than 12%	[74]
Phenolic acid	GC	Detects small phenolic acid compounds below 600 D	/
	HPLC + Magnetically assisted tube solid phase microextraction	Increased the analyte extraction efficiency from 44.9–64.0% to 78.6–87.1%	[86]
	RP-HPLC	Has good linearity and precision and can be used for commercial characterization	[87]
	Capillary electrophoresis	Hydrochloric acid concentration, 2 mol·L^−1^; temperature, 80 °C; and time 30 min; is a green extraction detection method	[88]
	UHPLC-DAD-ESI-QTOF-MS + UHPLC-DAD	Quantifies and characterizes phenolic compounds other than anthocyanins in blueberry juice and fermented wine	[89]

## Data Availability

Not applicable.

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
