# Peer review of "The Research Progress of Extraction, Purification and Analysis Methods of Phenolic Compounds from Blueberry: A Comprehensive Review"

_molecules, 2023, doi:10.3390/molecules28083610_

Round 1
Reviewer 1 Report
In my opinion Blueberry (vaccinium spp.) is a general name for blueberry plants of the genus Rhododendron?? I consider the topic not very orginal , it does not describe important active ingredients present in plants. I believe that this should be described in detail in the review paper The presented work does not bring anything new compared to other published materials. Authors should pay attention to proper citations, names of compounds, they write too chaotically. For example, they do not mention anything about tannins found in fruits and leaves. The chemical 88 structures of several major phenolic substances in blueberry shown in Figure 2 - different sizes of structures and captions under structures Mass spectrometry uses the principle of electromagnetism to ionize the substance to be 409 measured, and separates, detects and records the generated ions according to the size of 410 the mass-charge ratio. Becker L et al. used MALDI mass spectrometry imaging to simul- 411 taneously localize resveratrol, pterostilbene, and glucoside on grape leaves [64], which 412 was convenient and effective for the localization detection of pterostilbene. - description too general.Author Response
Thank you for your letter and for the reviewers’ comments concerning our manuscript entitled “The Research Progress of Extraction, Purification and Analysis methods of Phenolic compounds from blueberry: A Comprehensive Review” (Manuscript ID: molecules-2326858). Those comments are all valuable and very helpful for revising and improving our paper, as well as the important guiding significance to our researches. We appreciated very much the reviewers’ constructive and insight comments. We have studied comments carefully and have made correction which we hope meet with approval.
Revisions in the text are shown using yellow highlight for additions (causes excitotoxicity), and strikethrough font for deletions ().
The main corrections in the paper and the responds to the reviewer’s comments are as following:
In accordance with Reviewer 1’s suggestion:
Q1: In my opinion Blueberry (vaccinium spp.) is a general name for blueberry plants of the genus Rhododendron??
A1: Thanks for your suggestion. We have corrected the plant classification of blueberries: “Blueberry (Vaccinium spp.) belongs to the family Ericaceae, subfamily Vaccinoideae, genus Vaccinium [1].” We looked at the international plant names index and found the blueberry Life Science Identifier (LSID): urn:lsid:ipni.org:names:30000401-2.
Q2: I consider the topic not very origin, it does not describe important active ingredients present in plants. I believe that this should be described in detail in the review paper. The presented work does not bring anything new compared to other published materials.
A2: Thanks for your suggestion. According to your suggestion, we have added details about the ingredients of blueberry: “Blueberry fruit is greatly appreciated for its nutritional value of enriched organic acids (citric, ascorbic, phenolic acids and tannins), phenolic compounds (stilbenoids, tannins, and flavonoid compounds, including anthocyanin, flavanone, flavanol and quercetin), sugar, minerals, vitamins, fibers and pectins [3]. Phenolic compounds are divided into endogenous and exogenous phenols as per their source, and the ones present in plants are called endogenous phenols. By chemical composition, phenolic compounds can be classified into tannins, flavonoids and phenolic acids. Blueberries are rich in endogenous phenols, as the most beneficial components in blueberries, including anthocyanins, tannins, pterostilbene and phenolic acids.”
At present, there are a lot of researches on blueberry. But there is a lack of reviews on extraction, separation and purification techniques for blueberry. Compared with the previously published papers, this paper covers more comprehensive content. Efficient extraction and separation of components in blueberry is the key to development and utilization. Therefore, we summarized various extraction and separation methods of phenolic compounds in order to provide convenience for later research.
Q3: Authors should pay attention to proper citations, names of compounds, they write too chaotically. For example, they do not mention anything about tannins found in fruits and leaves.
A3: Thanks for your suggestion. We reclassified the compounds in the paper and unified their names. In addition, we have added a review of tannins in blueberries according to your suggestion. The details can be found in section 2.4Tannins.
“2.4 Tannins
2.4.1 Extraction methods
Solid-liquid extraction(SLE)
The principle of SLE is to use solvents to dissolve tannins in cells, commonly used solvents include water, ionic solvents, organic solvents, etc. Soxhlet device often used to extract the tannins in the dissolved in water and organic solvent [94]. Hagerman found that season, temperature, solvent and leaves with different treatments all affect-ed tannin extraction [95]. Durgawale, T.P.s found a combination of the following methods to extract tannins from wood and bark: Folin-Ciocalteu method for total phe-nols, vanillin (with sulfuric acid), or HCl/BuOH/Fe (â…¡) depolymerizing proanthocya-nidins, and Ate -Smith method for ellagtannins (or degradation in MeOH/HCl, then el-lagic acid if ellagtannin content is too low) [96]. Compared with this method, the new-er extraction methods such as ionic solvent methods are comparatively more efficient. Ionic liquids are mostly liquid organic salts. This kind of solvent has low vapor pres-sure, strong thermal stability and can dissolve a large number of substances, which greatly reduces the extraction time. But ionic solvents are too expensive to use on an industrial scale. Due to the high cost of ionic solvents, water is used in industry as the preferred solvent for SLE. Water can extract tannins at concentrations between 29 and 887 mg/g, and it produces residues without serious negative effects on the environ-ment. The tannin recovery rate of this solvent extraction is also very high, the only dis-advantage is that a large amount of solvent is required [97].
Supercritical Fluid Extraction (SFE)
Like the SFE we mentioned above the paper, there also used carbon dioxide as sol-vent. CO2 is a non-polar molecule, whereas tannins are considered polar compounds. The actual proportion of "solvent-soluble" and "solvent-resistant" tannins depends on the nature and concentration of the organic solvent [95]. Therefore, in order to improve the solubility of tannins and their extraction efficiency, polar co-solvents such as etha-nol, methanol, and water-based mixtures are often be added. This method of using CO2 as a solvent can create a non-toxic anaerobic environment. It can avoid strong oxi-dation reactions, and relatively mild temperatures do not destabilize tannins. The dis-advantage is the high cost of using high-pressure equipment, which is not suitable for large-scale industrial applications. Continuous mixing of solvent and particles also AIDS the extraction process, and better mixing results can be obtained when the ratio of solvent to solid is higher. However, this ratio can change the particle, but not the solvent of tannin extraction [98].
Microwave assisted extraction (MAE)
The method is a combination of traditional tannin extraction solvent and microwave rapid heating. According to the principle of solvent and material balance, the greater the amount of solvent, the greater the amount of tannin-10 extraction. In the work of Wang et al., it was found that an increase in solvent volume caused the extracted tan-nins to strengthen to a certain extent (ratio 1:30) and then begin to decline [99]. The main advantages of MAE are that compared with traditional solid-liquid extraction, the extraction time is greatly shortened and the solvent requirements are reduced. An-other advantage is that it improves mass transfer. The main disadvantages are the high cost of equipment required for mass production and the potential thermal degradation of raw materials and tannins [100].
Pressurized Water Extraction (PWE)
The method is based on the mass transfer properties of water to tannin at high tem-perature and pressure. The main difference from traditional SLE with hot water is that pressurized water extraction uses acceptable temperatures above the boiling point and pressures above atmospheric pressure to keep water in a liquid state. Tannins are ex-tracted under subcritical conditions (0.1-22 MPa, 100-374 °C) most of the time, but pressure and temperature also affect extraction. For example, when subcritical condi-tions are exceeded, extraction time will be shortened by 5-60 minutes [94]. Therefore, we can selectively extract different polar tannins by changing the temperature, pres-sure or solvent. However, due to the high cost, it is not suitable for large-scale indus-trial production.
Ultrasound Assisted Extraction (UAE)
Ana Rita Silva et al. compared HAE and UAE to determine which method was most suitable for promoting L from Cytinus hypocistis (L.). Subspecies of the genus Chry-santhus UAE has been shown have some advantages over HAE for intracellular tannin extraction, such as a shortened extraction time. In their studies, they found that while lower ethanol concentrations produced higher extraction rates (probably due to higher recovery of water-soluble carbohydrates), the highest levels of tannins were obtained using higher ethanol concentrations. In addition, the authors demonstrated that etha-nol/water mixtures (60:40, v/v) were superior to other organic solvents [101].
2.4.2 Separation and Purification
Tannins from crude plant extracts can be separated and purified by adsorption chro-matography. Dan, Liu. et al. further isolated and purified hydrolyzed tannins from ge-ranium crude extract by adsorption chromatography on a cross-linked 12% agarose gel (Superose 1210/300 GL). The purity of geranyl and geranyl extracted by this method were 92.4% and 87.2%, respectively, and the corresponding yields were 88.0 and 76.8%[102]. In addition, high speed countercurrent chromatography is often used to separate . Dan, Liu. et al. purified gallic acid, cilantro and geranin from 70% rutheni-um-iridium acetone aqueous extract by positive and reverse phase high speed counter-current chromatography. The solvent system used in this study was n-hexane - ethyl acetate - methanol-acetic acid - water (1: 10: 0: 0: 2: 20). The results show that tannin can be successfully separated by reversed-phase high speed counter-current chroma-tography and normal phase high speed counter-current chromatography [103].
2.4.3 Detection and Analysis
Chen Wang et al. used UHPLC-Q Exactive Orbitrap/MS positive and negative modes respectively, under optimized conditions (optimized conditions: ethanol concentration (pH = 3), 48%; Temperature, 50 °C, number of static cycles) to identify different types of polyphenols in blueberry extract. Eleven pro-anthocyanidins (tannins) in blueberry were identified by this method, including gallocatechin and epigallocatechin [104]. Annalisa Romani et al. analyzed and characterized several tannin extracts of myrtle and pomegranate using HPLC/DAD/ESI-MS method, and confirmed that water and water ethanol myrtle leaf extracts were rich in gallic side, gallic quinic acid, ellagtannin and flavonoids. The specific operating conditions of the mass spectrometer are: gas temperature 350 ° C, flow rate 10.0 L/min1, atomizer pressure 30 psi, tetrode tempera-ture 30 ° C and capillary voltage 3500 V. Monopolyphenols were quantified directly by HPLC/DAD using a five-point regression curve established by available criteria [105]. In addition, the tannin content of blueberries can be determined by ultraviolet spectro-photometry. The specific method is as follows: the 1:50 diluted sample and blank sam-ple were hydrolyzed with acid, and the absorbance was measured at 550nm. The re-sulting absorbance (sample blank) is then multiplied by a factor of 19.33 to calculate the total tannin concentration in g/L. These final values are expressed in mg / 100g dry matter (mg / 100g per minute) [106].”
Q4: The chemical 88 structures of several major phenolic substances in blueberry shown in Figure 2 - different sizes of structures and captions under structures.
A4: Thanks for your suggestion. We have corrected the mistakes. This is Figure 2 after adjusting the chemical formula and the size of the name.
Q5: Mass spectrometry uses the principle of electromagnetism to ionize the substance to be 409 measured, and separates, detects and records the generated ions according to the size of 410 the mass-charge ratio. - description too general.
A5: Thanks for your suggestion. We have described the principle and function of mass spectrometry detection in more detail. “Mass Spectrometry (MS) is a method that using electric and magnetic fields to separate moving ions according to their mass-to-charge ratio and then detecting the compounds. Through the analysis of the molecular weight of these ions can obtain compounds, chemical structure, cracking rule. This method has the characteristics of high accuracy, high sensitivity, especially suitable for separation of mixtures.”
Q6: Becker L et al. used MALDI mass spectrometry imaging to simul- 411 taneously localize resveratrol, pterostilbene, and glucoside on grape leaves [64], which 412 was convenient and effective for the localization detection of pterostilbene. - description too general.
A6: Thanks for your suggestion. We have revised the citation to this article to be more detailed, and the revised content is as follows: “Becker, L. et al. used MALDI-MS imaging to simultaneously localize resveratrol, pterostilbene, and glucoside on grape leaves [70]. Because MALDI is a soft ionization technique, molecular information can be retained and there is no need to label compounds of interest for detection, as with fluorescence microscopy. Therefore, it provides the method of "label free" imaging. It was convenient and effectively for the localization detection of pterostilbene.”
Reviewer 2 Report
The article fits the journal. However, there are a few revisions required:
a) English revision, try to summarize some sentences. Sometimes, the sentences are too long and extensive;
b) Same thing for the tables, in some case, they are very hard to read.
Besides, some of the good references out there on this topic are missing. For instance:
Zhang, Q., Cheng, Z., Wang, Y. and Fu, L., 2021. Dietary protein-phenolic interactions: Characterization, biochemical-physiological consequences, and potential food applications. Critical Reviews in Food Science and Nutrition, 61(21), pp.3589-3615.
Liu, J., Yong, H., Yao, X., Hu, H., Yun, D. and Xiao, L., 2019. Recent advances in phenolic–protein conjugates: Synthesis, characterization, biological activities and potential applications. RSC advances, 9(61), pp.35825-35840.
Liu, J., Pu, H., Liu, S., Kan, J. and Jin, C., 2017. Synthesis, characterization, bioactivity and potential application of phenolic acid grafted chitosan: A review. Carbohydrate Polymers, 174, pp.999-1017.
Cheung, L.K., Sanders, A.D., Pratap-Singh, A., Dee, D.R., Dupuis, J.H., Baldelli, A. and Yada, R.Y., 2023. Effects of high pressure on protein stability, structure, and function—Theory and applications. In Effect of High-Pressure Technologies on Enzymes (pp. 19-48). Academic Press.
Author Response
[Response List of Molecules Journal]
2023.04.15
Re: Manuscript ID: molecules-2326858
Dear Editors and Reviewers:
Thank you for your letter and for the reviewers’ comments concerning our manuscript entitled “The Research Progress of Extraction, Purification and Analysis methods of Phenolic compounds from blueberry: A Comprehensive Review” (Manuscript ID: molecules-2326858). Those comments are all valuable and very helpful for revising and improving our paper, as well as the important guiding significance to our researches. We appreciated very much the reviewers’ constructive and insight comments. We have studied comments carefully and have made correction which we hope meet with approval.
Revisions in the text are shown using yellow highlight for additions (causes excitotoxicity), and strikethrough font for deletions ().
The main corrections in the paper and the responds to the reviewer’s comments are as following:
In accordance with Reviewer 2’s suggestion:
Q1: English revision, try to summarize some sentences. Sometimes, the sentences are too long and extensive.
A1: Thanks for your suggestion. According to your suggestion, our article has been polished by native English speakers.
Q2: Same thing for the tables, in some case, they are very hard to read.
A2: Thanks for your suggestion. We changed the expressions in the table to make it more concise.
Table 2.The separation and purification methods of phenolic compounds from blueberry were summarized in this paper
|
Phenolic Compound |
Separation and purification Methods |
Separation and purification Condition |
Purity or Characteristic |
Ref. |
|
|
|
Anthocyanin |
HPLC method |
Acetonitrile-water (containing 0.3% phosphoric acid) as the mobile phase gradient elution at 520 nm detection wavelength
|
99% |
[44] |
|
|
|
Semi-preparative high performance liquid chromatography |
Mobile phase A:methanol B:3% for-mic acid. The initial gradient composition: 15% solvent A and 85% solvent B. The elution conditions: sol- vent B: 0min, 85%; 3min, 80%; 7min, 75%; 10min, 75%; 55min, 30%; 60 min, 30%; 65 min.
|
Three anthocyanin components purity are 97.7%, 99.3%, and 95.4% |
[45] |
|||
|
HSCCC |
Duplexic mixture of tert-butyl methyl ether/n-butanol/acetonitrile/water (2:2:1:5) acidified with trifluoroacetic acid
|
Peak purity standard |
[46] |
|||
|
Macroporous resin method |
Macroporous resin combined with Sephadex LH-20 method
|
90.96% |
[48] |
|||
|
Membrane separation method |
Use nanofiltration membranes (NF245 and NF270) to separation and adsorption of anthocyanins
|
Reduce anthocyanin waste by more than 60% |
[51] |
|||
|
DSC-C18+DSC-SCX+HPLC |
Hydrophobic silica gel (DSC-C18) and cation exchange resin (DSC-SCX) in two consecutive solid-phase extractions anthocyanins |
100% |
[52] |
|||
|
Pterostilbene |
New ways to increase solubility |
PTS:BCD:HPMC ternary system using ethanol as a co-solvent |
low-bulk powders with a high content of PTS |
[68],[69] |
||
|
Phenolic acid |
UPC |
Mobile phase: Carbon dioxide, methanol/acetonitrile (00:70,v/v) Flow rates of 30.1 mL/min. Modifier:17% TFA. The shortest time:0.8 min. |
A short separation time of 0.8min |
[86] |
||
Table 3. Summary of the detection methods of phenolic compounds in blueberries involved in this paper
|
Phenolic Compound |
Detection Method |
Characteristic |
Ref. |
|
|
|
Anthocyanin |
UV-VIS |
Maximum absorption in the visible light range around 520 nm |
[55] |
|
|
|
HPLC |
Chromatographic fractionation that methanol media achieved suitable chromatographic performance |
[56] |
|||
|
Mass spectrometry |
Q-TOF and ESI are the most commonly used |
[55] |
|||
|
HPLC-DAD+MSn |
Extracted pigments from berry peels with 0.1% HCl methanol, purified using C-18 solid phase columns |
[57] |
|||
|
UV-VIS+LC-MS |
Provide additional information on the structural details of anthocyanins |
[60] |
|||
|
UPLC/UV/MS |
The mobile phase A: water; B:acetonitrile). Each containing 0.1% formic acid. With a flow rate of 0.3 mL/min, this method provides a reliable determination of anthocyanins. |
[61] |
|||
|
HPLC+MSn |
Distinguish anthocyanins from pro-anthocyanidins |
[63] |
|||
|
Pterostilbene |
MALDI mass spectrometry |
Simultaneously localize resveratrol, pterostilbene, and glucoside on grape leaves
|
[70] |
||
|
RP-HPLC |
The addition of 0 mM HP-β-CD to a 25:7 (v/v) methanol aqueous mobile phase at 12°C and pH 50.50 significantly improved the main analytical parameters
|
[73] |
|||
|
DLLME |
A linear response:5000 ng/mL (-1). Accuracy: overall recovery of 1000% to 1% for samples spiked at different levels of 90 to 102 ng/mL (-50) with a standard deviation of less than 12% |
[74] |
|||
|
Phenolic acid |
GC |
Detect small phenolic acid compounds below 600D |
/ |
||
|
HPLC+ Magnetically assisted tube solid phase microextraction |
Increased the analyte extraction efficiency from 44.9-64.0% to 78.6-87.1% |
[89] |
|||
|
RP-HPLC |
Have good linearity and precision, and can be used for commercial characterization |
[90] |
|||
|
Capillary electrophoresis |
Hydrochloric acid concentration (2 mol· L-1), temperature (80°C) and time (30 minutes),is a green extraction detection method |
[91] |
|||
|
UHPLC-DAD-ESI-QTOF-MS+UHPLC-DAD |
Quantify and characterize phenolic compounds other than anthocyanins in blueberry juice and fermented wine |
[92] |
|||
Q3: Besides, some of the good references out there on this topic are missing. For instance:
Zhang, Q., Cheng, Z., Wang, Y. and Fu, L., 2021. Dietary protein-phenolic interactions: Characterization, biochemical-physiological consequences, and potential food applications. Critical Reviews in Food Science and Nutrition, 61(21), pp.3589-3615.
Liu, J., Yong, H., Yao, X., Hu, H., Yun, D. and Xiao, L., 2019. Recent advances in phenolic–protein conjugates: Synthesis, characterization, biological activities and potential applications. RSC advances, 9(61), pp.35825-35840.
Liu, J., Pu, H., Liu, S., Kan, J. and Jin, C., 2017. Synthesis, characterization, bioactivity and potential application of phenolic acid grafted chitosan: A review. Carbohydrate Polymers, 174, pp.999-1017.
Cheung, L.K., Sanders, A.D., Pratap-Singh, A., Dee, D.R., Dupuis, J.H., Baldelli, A. and Yada, R.Y., 2023. Effects of high pressure on protein stability, structure, and function—Theory and applications. In Effect of High-Pressure Technologies on Enzymes (pp. 19-48). Academic Press.
A3: Thanks for your suggestion. We have downloaded those articles and read it carefully. These articles are very useful for our review. We have cited these documents in the paper (Reference 4,67,85,114).
We tried our best to improve the manuscript and we appreciate for Editors/Reviewers’ warm work earnestly and hope that these revisions are satisfactory and that the revised version will be acceptable for publication in Molecules.
Once again, thank you very much for your comments and suggestions.
Yours sincerely,
Dr. Ronghua Fan
Address: School of Public Health, Shenyang Medical College, No.146 Huanghe North Street, Shenyang 110000, PR China
Ph: +86 13516035668;
E-mail address: rh_fan@163.com
Reviewer 3 Report
This review article systematically describes methods for the extraction, purification, detection, and characterization of three phenolic compounds (anthocyanines, pterostilbene, and phenolic acids) from blueberries. Although new and insightful perspectives were not provided, this paper provided a good compilation of a lot of data on extraction, purification, detection methods of some phenolic compounds in blueberries. This paper is recommended for publication in Molecules. However, please respond appropriately to the following points.
The sub-headings are not appropriate and not systematically numbered, please correct them as follows.
(line 102) 2. The Research Progress of extraction, purification and detection methods of Anthocyanins → 2. The Research Progress of Extraction, Purification and Detection Methods of Some Phenolic Compounds from Blueberries
2.1. Anthocyanines
(line 104) 2.1. Extraction methods and progress → 2.1.1. Extraction methods
(line 121) Solvent Extraction Method → Solvent extraction
(line 174) Deep eutectic solvent and its extension → Extraction using deep eutectic solvent
(line 192) Enzyme assisted extraction method → Enzyme assisted extraction
(line 205) Supercritical Carbon Dioxide Extraction (SCDE) → Extraction using supercritical fluid carbon dioxide
(line 240) 2.2. Separation and purification of anthocyanin → 2.1.2. Separation and purification
(line 251) Chromatographic method → High performance liquid chromatography (HPLC)
(line 276) Column chromatography method → Column chromatography
(line 293) Membrane separation method → Membrane separation
(line 310) 2.3 Detection and analysis methods of anthocyanins → 2.1.3 Detection and analysis methods
(line 371) 3. Pterostilbene → 2.2. Pterostilbene
(line 372) 3.1 Extraction of pterostilbene → 2.2.1. Extraction methods
(line 391) 3.2 Isolation and purification of pterostilbene → 2.2.2. Separation and purification
(line 407) 3.3 Detection of pterostilbene → 2.2.3 Detection and analysis methods
(line 450) 4. Phenolic acids → 2.3. Phenolic acids
(line 451) 4.1 Extraction method and progress of phenolic acid → 2.3.1. Extraction methods
(line 484) 4.2 Separation and purification of phenolic acids → 2.3.2. Separation and purification
(line 502) 4.3 Analytical detection methods of phenolic acid. → 2.3.3. Detection and analysis methods
(line 555) 5. Outlook → 3. Outlook
(line 556) 5.1 Food value and prospects of blueberries → 3.1. Food value and prospects of blueberries
(line 575) 5.2 Health value and prospects of blueberries → 3.2 Health value and prospects of blueberries
(line 609) 6. Conclusions → 4. Conclusions
Author Response
[Response List of Molecules Journal]
2023.04.15
Re: Manuscript ID: molecules-2326858
Dear Editors and Reviewers:
Thank you for your letter and for the reviewers’ comments concerning our manuscript entitled “The Research Progress of Extraction, Purification and Analysis methods of Phenolic compounds from blueberry: A Comprehensive Review” (Manuscript ID: molecules-2326858). Those comments are all valuable and very helpful for revising and improving our paper, as well as the important guiding significance to our researches. We appreciated very much the reviewers’ constructive and insight comments. We have studied comments carefully and have made correction which we hope meet with approval.
Revisions in the text are shown using yellow highlight for additions (causes excitotoxicity), and strikethrough font for deletions ().
The main corrections in the paper and the responds to the reviewer’s comments are as following:
In accordance with Reviewer 3’s suggestion:
Q1: This paper is recommended for publication in Molecules. However, please respond appropriately to the following points.
The sub-headings are not appropriate and not systematically numbered, please correct them as follows.
(line 102) 2. The Research Progress of extraction, purification and detection methods of Anthocyanins → 2. The Research Progress of Extraction, Purification and Detection Methods of Some Phenolic Compounds from Blueberries
2.1. Anthocyanins
(line 104) 2.1. Extraction methods and progress → 2.1.1. Extraction methods
(line 121) Solvent Extraction Method → Solvent extraction
(line 174) Deep eutectic solvent and its extension → Extraction using deep eutectic solvent
(line 192) Enzyme assisted extraction method → Enzyme assisted extraction
(line 205) Supercritical Carbon Dioxide Extraction (SCDE) → Extraction using supercritical fluid carbon dioxide
(line 240) 2.2. Separation and purification of anthocyanin → 2.1.2. Separation and purification
(line 251) Chromatographic method → High performance liquid chromatography (HPLC)
(line 276) Column chromatography method → Column chromatography
(line 293) Membrane separation method → Membrane separation
(line 310) 2.3 Detection and analysis methods of anthocyanins → 2.1.3 Detection and analysis methods
(line 371) 3. Pterostilbene → 2.2. Pterostilbene
(line 372) 3.1 Extraction of pterostilbene → 2.2.1. Extraction methods
(line 391) 3.2 Isolation and purification of pterostilbene → 2.2.2. Separation and purification
(line 407) 3.3 Detection of pterostilbene → 2.2.3 Detection and analysis methods
(line 450) 4. Phenolic acids → 2.3. Phenolic acids
(line 451) 4.1 Extraction method and progress of phenolic acid → 2.3.1. Extraction methods
(line 484) 4.2 Separation and purification of phenolic acids → 2.3.2. Separation and purification
(line 502) 4.3 Analytical detection methods of phenolic acid. → 2.3.3. Detection and analysis methods
(line 555) 5. Outlook → 3. Outlook
(line 556) 5.1 Food value and prospects of blueberries → 3.1. Food value and prospects of blueberries
(line 575) 5.2 Health value and prospects of blueberries → 3.2 Health value and prospects of blueberries
(line 609) 6. Conclusions → 4. Conclusions
A1: Thanks for your helpful comments. All the subheadings have been revised in the paper according to your requirements. Thank you very much for your care and patience.
We tried our best to improve the manuscript and we appreciate for Editors/Reviewers’ warm work earnestly and hope that these revisions are satisfactory and that the revised version will be acceptable for publication in Molecules.
Once again, thank you very much for your comments and suggestions.
Yours sincerely,
Dr. Ronghua Fan
Address: School of Public Health, Shenyang Medical College, No.146 Huanghe North Street, Shenyang 110000, PR China
Ph: +86 13516035668;
E-mail address: rh_fan@163.com
Round 2
Reviewer 1 Report
I think that manuscript can be accept in the present form.